# How Do Plants Cope with DNA Damage? A Concise Review on the DDR Pathway in Plants

**DOI:** 10.3390/ijms24032404

**Published:** 2023-01-26

**Authors:** Miriam Szurman-Zubrzycka, Paulina Jędrzejek, Iwona Szarejko

**Affiliations:** Institute of Biology, Biotechnology, and Environmental Protection, Faculty of Natural Sciences, University of Silesia, 40-032 Katowice, Poland

**Keywords:** DDR, DNA damage, DNA repair, cell cycle, plants

## Abstract

DNA damage is induced by many factors, some of which naturally occur in the environment. Because of their sessile nature, plants are especially exposed to unfavorable conditions causing DNA damage. In response to this damage, the DDR (DNA damage response) pathway is activated. This pathway is highly conserved between eukaryotes; however, there are some plant-specific DDR elements, such as SOG1—a transcription factor that is a central DDR regulator in plants. In general, DDR signaling activates transcriptional and epigenetic regulators that orchestrate the cell cycle arrest and DNA repair mechanisms upon DNA damage. The cell cycle halts to give the cell time to repair damaged DNA before replication. If the repair is successful, the cell cycle is reactivated. However, if the DNA repair mechanisms fail and DNA lesions accumulate, the cell enters the apoptotic pathway. Thereby the proper maintenance of DDR is crucial for plants to survive. It is particularly important for agronomically important species because exposure to environmental stresses causing DNA damage leads to growth inhibition and yield reduction. Thereby, gaining knowledge regarding the DDR pathway in crops may have a huge agronomic impact—it may be useful in breeding new cultivars more tolerant to such stresses. In this review, we characterize different genotoxic agents and their mode of action, describe DDR activation and signaling and summarize DNA repair mechanisms in plants.

## 1. Introduction

All living organisms are continuously exposed to various environmental stresses, some of which may cause DNA damage, either directly or indirectly (e.g., by increasing the level of reactive oxygen species—ROS). Due to their immobile nature, plants are not able to avoid these stresses. Moreover, cell metabolism itself may also introduce DNA lesions through, e.g., replication errors or ROS production. Therefore, proper sensing of DNA damage and precise activation and functioning of the DNA repair machinery is of great importance in preserving genome integrity. The signal of DNA damage is processed through pathways collectively termed DDR (DNA damage response), which form a complex multilevel signaling network leading to the activation of processes related to DNA repair. Simultaneously, in the case of proliferating cells, DDR leads to cell cycle arrest, through control checkpoints, for as long as the DNA is not repaired. Various genotoxic agents causing different DNA lesions have been used to study DDR in plants. It is well known that distinct DNA repair mechanisms are activated depending on the type of DNA damage. The aim of this review is to summarize the current knowledge regarding the DNA damage response pathway in plants.

## 2. Wide Range of Genotoxic Agents Activating DDR in Plants

The activity of genotoxic factors leads to the generation and accumulation of DNA damage, which triggers the DDR pathway. Unrepaired or not correctly repaired DNA damage gives rise to mutations that can lead to many abnormalities in the cell metabolism or even cell death. There are diverse types of DNA damage that can be grouped into two main categories: (1) disturbing only one DNA strand and (2) affecting both DNA strands. The first group includes lesions such as single-strand breaks (SSBs), oxidized/alkylated bases, DNA adducts, intra-strand crosslinks (CLs) between adjacent guanines, and DNA photoadducts. The second group includes inter-strand CLs between opposite DNA strands, and double-strand breaks (DSBs) (Figure 1) [1]. Among them, DSBs are the most harmful because they can lead to the loss of genetic material through chromosomal fragmentation, and as a consequence, they can lead to cell death [2,3].

Diverse groups of exogenous genotoxic agents causing different types of DNA damage can be used to study distinct elements of the DDR pathway and DNA repair mechanisms. One of the groups of chemical genotoxins comprises alkylating agents. The main targets of monofunctional alkylators are guanines and adenines that can be methylated at O and N positions, leading to the generation of DNA lesions such as O^6^-methylguanine N^7^-methylguanine, N^3^-methyladenine, or others [4,5]. The alkylated base is not appropriately recognized during replication, which leads to mispairing with incorrect nucleotides. It can also be removed (base loss), which creates an abasic site (AP-Apurinic/APyrimidinic site) [6]. Alkylating agents, such as EMS (Ethyl MethaneSulfonate), MMS (Methyl MethaneSulphonate), or MNU (N-Methyl-N-NitrosoUrea), are used to study DDR, but they are also widely used to induce mutations for plant breeding and reverse genetic studies, e.g., through the creation of TILLING populations [7,8,9,10,11,12,13,14,15]. Apart from monofunctional alkylators, there are also bi- and polyfunctional ones, such as nitrogen mustards or platinum drugs. They damage DNA by generating bulky adducts to nucleotides, intra-strand crosslinks, as well as inter-strand crosslinks that block DNA-related processes, such as replication and transcription. Because of their ability to block these processes in tumor cells with a high proliferation rate, they are widely used in cancer chemotherapy [16,17,18]. In plant studies, the most frequently used DNA crosslinking agents are cisplatin (cis-diamminedichloroplatinum II, CDDP) and mitomycin C [19,20,21,22,23,24,25].

Another group of chemical genotoxins includes radiomimetics, whose name indicates that they affect DNA similarly to ionizing radiation. They induce double-strand breaks through oxidative damage. The most common radiomimetic used to generate DSBs in plants’ genomes is bleomycin (BLM) [26,27,28,29]. This glycopeptidic antibiotic isolated from *Streptococcus verticillus* is also used as an antitumor agent. It is known to directly produce ROS by forming a complex with molecular oxygen and divalent ions such as iron. The bleomycin/iron complex binds to the DNA helix through a bithiazol ring, which causes a DNA strand scission and lipid peroxidation [18,30]. Zeocin is another radiomimetic antibiotic that belongs to the bleomycin family and has recently been used most often to study DDR pathways in plants [31,32,33,34].

Genotoxic stress can also be induced in plant cells by other chemical agents, such as hydroxyurea (HU), which acts at the S-phase of the cell cycle, as it is a ribonucleotide reductase (RNR) inhibitor. RNR is an enzyme crucial for the formation of deoxyribonucleotides—it induces the conversion of ribonucleotides to deoxyribonucleotides by catalyzing the removal of a 2′OH group from ribonucleoside diphosphates. Therefore HU causes DNA replication stoppage. HU has been used in hematological disorders and cancer therapy as an antiproliferative drug since the 1960s [35,36]. Its action is easily reversible; therefore, short HU treatment is often used in scientific research to synchronize the cell cycle. However, higher concentrations and more prolonged treatment may lead to the accumulation of DNA damage [37]. Camptothecin (CPT) is another genotoxic factor that has been used to study DDR in plants, e.g., in *Medicago truncatula* [38]. Its cytotoxic effects have been long known for plants [39]. CPT causes the induction of both SSBs and DSBs and enhances the level of cell death in plants [40,41]. It is also widely used in anticancer therapies—it inhibits the activity of topoisomerase I (TopI), which is an enzyme required for the relaxation of DNA supercoiling after replication, transcription, etc. CPT acts through intercalating between DNA breaks flanking the TopI–cleavage complex [42,43]. Among the chemical factors used for the creation of DNA damage in plants is also zebularine, which is a cytidine analog leading to the formation of DNA–protein crosslinks. It has been recently used to study DDR, i.e., in Arabidopsis [44].

There are two main physical factors causing DNA damage: ionizing and non-ionizing radiation. The most common physical agent used in plant DDR studies and mutation breeding is ionizing radiation (IR) with, e.g., γ-rays, X-rays, or ion beams. IR is known to induce DNA damage directly by ionizing DNA molecules, which may cause DSBs and hence DNA fragmentation. It also acts indirectly through the radiolysis of water that leads to ROS production, e.g., hydroxyl and hydrogen radicals (OH● and H●) and free electrons (e^−^). The most frequent DNA lesions caused by thus-induced oxidative stress are oxidized bases (e.g., thymine glycols and 8-oxoguanine), base loss (abasic sites), and SSBs. Among IR-induced DNA damages are also DNA–protein crosslinks [1,45]. The non-ionizing radiation also leads to DNA damage. Plants need sunlight for photosynthesis and survival. However, the UV light that is a part of solar energy is harmful to DNA (UV-B in particular). It generates photoadducts, mainly cyclobutene pyrimidine dimers (CPDs) and pyrimidine (6-4) pyrimidone dimers (6-4PPs). In CPDs, there are covalent bonds between the C-5 and C-6 carbon atoms of neighboring pyrimidines (mainly TT, less often TC and CC), forming a four-member ring structure. In 6-4PPs, there is a bond between the C-6 and C-4 carbon atoms in TC dinucleotide. The proportion and distribution of these photoadducts in the genome depend on the nucleotide composition of DNA and chromatin structure [1,46]. However, it has been estimated that in plants, CPDs are the major types of UV-induced DNA lesions (approximately 75%, up to 90%) [47,48]. Both CPDs and 6-4PPs can block the transcription and replication processes. The replication blockage can lead to the collapse of replication forks that can generate DSBs [49,50]. Additionally, UV radiation can induce ROS production and oxidative damage (pyrimidine hydrates) and DNA–DNA as well as DNA–protein crosslinks [48].

UV light is the most ubiquitous genome-damaging factor worldwide, but there are also other environmental stresses, which plants are constantly exposed to, that may damage DNA. Abiotic stresses such as drought, heat, cold, or soil salinity are known to promote ROS formation and accumulation, which results in oxidative stress that can cause DNA damage [51,52]. For example, it is long known that cold stress induces chromatin fragmentation and apoptotic changes in tobacco [53]. Later work on Arabidopsis treated with low temperature (4 °C) has shown that chilling stress provokes DNA damage mainly in root stem cells and their early descendants [54]. On the other hand, DNA integrity may also be disrupted by heat stress that triggers nucleotide modifications and single-strand as well as double-strand breaks, and changes in chromatin architecture [55,56,57]. Salinity stress has also been proven to induce DNA strand breaks, e.g., in Arabidopsis [58] and rice [59], predominantly through the production of ROS. The application of ROS-specific antioxidants significantly reduces the amount of DNA breaks caused by NaCl [60,61,62]. Much evidence shows that heavy metals, such as cadmium (Cd) and lead (Pb), cause oxidative and genotoxic stress [63]. Aluminum (Al), the most common metal in the Earth’s crust, is highly phytotoxic in acidic soils (at pH below 5.5), which now comprise more than 50% of arable lands. Under acidic conditions, Al is known to affect genome integrity, induce DSBs in root meristems and activate the DDR pathway [64,65,66]. It is worth mentioning that low pH itself induces oxidative stress in plant cells and hence, through the production of ROS, it may also lead to DNA damage [67]. So, plants are continuously exposed to conditions harmful to their DNA. It has to be noted that DNA damage can also be induced by endogenous factors that are the products of cell metabolism. All processes associated with DNA, such as replication, recombination, etc., are not free of mistakes and can introduce DNA lesions. It has been estimated that the DNA of each living organism accumulates thousands of lesions every single day [68]. Thus, the importance of DDR mechanisms in maintaining genome integrity and stability cannot be overestimated. Exploring the details of the DDR pathways may be essential for fully understanding plant responses to environmental stresses and breeding new cultivars tolerant to them.

## 3. DDR–Sensing and Signaling the DNA Damage

Cells are constantly subjected to DNA damage and all organisms, including plants, have evolved efficient mechanisms for sensing this damage (Figure 2). Two key factors involved in the recognition of DNA lesions are ATM (Ataxia Telangiectasia Mutated) and ATR (ATM and Rad3-related), protein kinases belonging to the phosphatidylinositol 3-kinase-like family. In principle, they play distinct and additive roles—ATM recognizes DSBs, whereas ATR is predominantly recruited to ssDNA and stalled replication forks—the hallmarks of replication stress [69]. These kinases are activated by different DNA damage sensors. Similarly, as in the case of animals and yeast, in plants, ATM is activated by the MRN complex (MRE11-RAD50-NBS1) that senses DSBs [70]. ATM is targeted to the DSBs sites by the C-terminus of NBS1 (NIJMEGEN BREAKAGE SYNDROME 1) [71]. ATR is recruited to ssDNA by a different mechanism. ATR is inactive in complex with ATRIP (ATR interacting protein) and is recruited to ssDNA through interaction with RPA (replication protein A) that senses and coats ssDNA. In parallel, another group of proteins is also recruited by RPA to the ssDNA site—a group that comprises DNA polymerase α, RFC (RAD17-replication factor C), and 9-1-1 complex (RAD9, RAD1, and HUS1). This complex is involved in ATR activation [72]. The DDR pathway has been extensively studied in animals because its malfunction is related to cancer development. Nevertheless, in plant research, it has also been studied in detail for the last 20 years. The elements involved in sensing DNA damage are highly conserved among all eukaryotes. Arabidopsis homologs of ATM and ATR kinases were identified in the early 2000s [73,74], and plant homologs of DNA damage sensors mentioned above were identified shortly thereafter [75,76,77].

The activated ATM and ATR kinases start a signaling cascade and phosphorylate a plethora of downstream elements of the DDR pathway. One of these elements is γH2AX (a histone variant phosphorylated at Ser-139 residue) that is accumulated at DNA damage sites in an ATM/ATR-dependent manner [70]. γH2AX is commonly known as a sensitive marker of DNA damage [78]. A recently identified XIP protein (γH2AX-INTERACTING PROTEIN) was found to interact directly with γH2AX, as well as with RAD51, the key recombinase involved DNA repair through HR (homologous recombination) [79]. Both ATM and ATR kinases phosphorylate and activate SOG1 (Suppressor Of Gamma 1)—the master regulator of DDR [80,81]. SOG1 was first identified from a gamma radiation-induced DNA damage suppressor screen in Arabidopsis (where *uvh1*–UV hypersensitive mutant was mutagenized) [82]. SOG1 is a transcription factor belonging to the NAC (NAM, ATAF1/2, and CUC2) family, which is specific to plants. It is considered to be a functional equivalent of p53—a tumor suppressor that controls DNA repair and cell cycle stoppage in response to DNA damage in animals. However, p53 and SOG1 are not true homologs, because they do not share any sequence similarity [83]. SOG1 is activated via phosphorylation of conserved C-terminal serine-glutamine (SQ) motifs [80]. Thus-activated SOG1 orchestrates the mechanisms of DNA repair, cell cycle arrest, endoreduplication, and PCD through transcriptional regulation [84]. An enormous number of genes were found to be directly or indirectly controlled by SOG1; e.g., ChIP-seq analysis in Arabidopsis revealed that more than 140 genes are its direct targets [85]. Very sophisticated DDR transcriptional models have been created by Bourbousse et al., confirming that SOG1 is a major activator of genes upregulated upon DNA damage. It was calculated that SOG1 directly controls ~8% of the transcriptional response to DNA damage caused by γ-radiation in Arabidopsis [86]. It is confirmed by many studies that SOG1 is a direct positive regulator of genes related to DNA repair (e.g., *RAD51* and *BRCA1* involved in HR or *PARP1* and *PARP2* involved in NHEJ), but also of the genes required for the cell cycle arrest (such as *SMR5* and *SMR7*–*SIAMESE-RELATED* cyclin-dependent kinase inhibitors) [81,85,86,87]. SOG1 is also known to induce the expression of other transcription factors, belonging to different TF families (i.e., WRKY or Zn Finger), including its own homologs from the NAC family, such as *NAC044* and *NAC085* [86].

Recent studies have revealed additional pathways of DDR in plants. The new player, whose function depends on ATM/ATR activity, is RBR1 (RETINOBLASTOMA-RELATED1), which is the only Arabidopsis homolog of animal tumor suppressor pRb. However, RBR was not found to be a direct target of ATM/ATR, hence the process of its activation upon DNA stress remains to be elucidated [88]. In general, RBR1 forms complexes with E2F transcription factors that are regulators of genes involved in entrance into the S phase during the cell cycle. The E2F transcription factors activate the expression of S-phase genes, and the RBR1 inhibits the E2F activity [89]. Upon genotoxic stress, RBR1, together with its binding partner E2FA, was found to accumulate at damage sites in nuclei and, together with BRCA1, it was involved in the recruitment of the DNA repair machinery [90,91]. Importantly, RBR1/E2FA complex regulates the expression of many DDR genes (e.g., *WEE1*) by association with their promoters, in a SOG1-independent manner [92].

## 4. DNA Repair Mechanisms

In principle, when a DNA lesion occurs it is important to repair it before replication and cell division. Numerous different mechanisms of DNA repair may be activated, depending on the type of DNA damage. DSBs may be repaired through homologous recombination (HR) or non-homologous end-joining (NHEJ); SSBs, AP sites, and some alkylated bases may be fixed by base excision repair (BER); changes caused by UV radiation (such as 6-4PP, CPD) may be corrected through direct reversal repair or nucleotide excision repair (NER); and polymerase mistakes may be removed by the mismatch repair mechanism (MMR) reviewed in [1,93,94,95]. These mechanisms are highly conserved among all eukaryotes.

### 4.1. Repair Mechanisms of Damage in Single DNA Strand

If damage occurs in one DNA strand only, the sequence information from the second, complementary strand may be used for its repair. The three most common pathways for the repair of these types of damages are BER, NER, and MMR.

BER (base excision repair) is triggered by a broad range of DNA lesions—by damaged or modified (alkylated, oxidized, or deaminated) DNA bases [96]. BER is initiated by DNA glycosylases that remove the damaged DNA bases and create AP sites. The AP sites may be formed after the accumulation of uracil in the DNA strand, which is caused by hydrolytic deamination of 5-methylcytosine. As a result, the U-G pairs are created, which, if not repaired before replication, will lead to C/G to T/A mutations. In this case, the BER mechanism is activated by UDG (URACIL DNA GLYCOSYLASE), which cuts the N-glycosidic bond and then cleaves the uracil out, leading to the creation of AP sites [97,98]. In Arabidopsis, there are several enzymes from the uracil-DNA glycosylase family involved in the removal of uracil from DNA [99,100]. In the case of other damaged or modified bases, the proper lesion-specific DNA glycosylases initiate BER by removing the affected base and thus generating the AP site. The first identified glycosylase involved in the removal of alkylated DNA bases in plants was 3-METHYLADENINE-DNA GLYCOSYLASE [101]. AP sites are then processed by AP endonucleases and/or AP lyases, which cleave the sugar-phosphate backbone at the AP site. The AP lyase activity is associated with bifunctional DNA glycosylases. As it follows, during the BER pathway the single-strand breaks are created, which may also be substrates for other repair mechanisms, such as NER [102]. The next steps of BER, following the base removal and AP site incision, include cleaning of DNA termini, and filling the gap by proper polymerases. Depending on the number of nucleotides incorporated, there are two types of BER–short-patch (SP-BER) with insertion of only one nucleotide and long-patch (LP-BER) with insertion of several (usually 2–13) nucleotides. In mammals, different polymerases are used in SP-BER and LP-BER (pol β and pol δ/ε, respectively) [103,104]. The last step of BER is DNA nick ligation. In mammals, during SP-BER, the ligation is proceeded by a complex consisting of XRCC1 and LigIIIa, whereas in LP-BER by LIG1 [104,105]. Plants possess the orthologs of most BER genes found in other kingdoms, however with some exceptions; e.g., plants do not have orthologs of pol β or LigIII related to SP-BER in animals. Additionally, some plant-specific BER proteins have also been identified, which indicates that some plant-specific characteristics emerged along BER evolution reviewed in [106]. Both mono- and bifunctional glycosylases have been found in Arabidopsis. Among bifunctional ones, with glycosylase and lyase activity, are, e.g., AtFPG and AtOGG1. Together with ZDP 3′ DNA phosphatase and ARP endonuclease, they are confirmed to be involved in the repair of oxidized bases [107]. Recently, it was shown that APE2 endonuclease and ZDP phosphatase play overlapping roles in maintenance of epigenome and genome stability in Arabidopsis [108]. Initially, because of the lack of plant orthologs of pol β and LigIII, which are the main players involved in SP-BER in mammals, it was assumed that plants do not repair damaged bases through this subpathway of BER [109]. However, it has been confirmed that in Arabidopsis uracil and AP sites may be repaired by both LP- as well as SP-BER [81,82]. Plant Pol λ, which belongs to the same family as Pol β (family X), is probably implicated in the synthesis step of the BER pathway in Arabidopsis [99,110]. On the other hand, plants possess orthologs of Pol δ and ε, which are involved in LP-BER in mammals. Nevertheless, their potential function in LP-BER needs to be further investigated [106]. Arabidopsis genome encodes three ligases—AtLIG1, AtLIG4, and AtLIG6. The last one is plant-specific and together with AtLIG4 is critical for seed longevity [111]. AtLIG4 is confirmed to be implicated in DSB repair [112,113]. AtLIG1 is the only ligase known to be responsible for the final ligation of DNA ends in both BER subpathways (SP- and LP-) in Arabidopsis [100]. AtXRCC1 stimulates 3′-end cleaning by ZDP and enhances the ligation step, probably by interaction with AtLIG1 [114]. In plants, the proper functioning of the BER pathway is crucial for seed longevity, since seed storage leads to the accumulation of oxidative changes, and BER is involved in repairing oxidative DNA lesions in germinating embryos [107,115]. The list of BER-related proteins in plants is provided in Table 1.

NER (nucleotide excision repair) is used to repair various bulky DNA adducts, such as UV-induced photoproducts, that cause helix distortion in the DNA structure. In humans, deficiencies in NER-related genes lead to several disorders, e.g., xeroderma pigmentosum. Interestingly, NER-related proteins were found to be more conserved among bacteria, yeast, plant, and animal species than proteins from other DNA repair pathways [116]. There are two NER subpathways in mammals: (1) global genomic repair (GGR or GG-NER) and (2) transcriptional-coupled repair (TCR or TC-NER), which differ in the mode of damage recognition, but use the same machinery to correct the lesions [94,117]. In the GG-NER subpathway, damage that occurs anywhere in the genome may be detected. In animals, the repair is initiated by the XPC-HR23B-CEN2 (Xeroderma Pigmentosum group C-Homolog of Rad23B–Centrin2) complex. This complex scans DNA and detects helix distortion. This process is very often enhanced by the activity of the DDB complex (damaged DNA-binding complex), composed of DDB1 and DDB2 subunits, which helps to find lesions that only slightly alter the structure of DNA, such as CPDs. In the TC-NER, only the damage that occurs in the transcribed DNA strand of highly expressed genes is detected. The repair is initiated by RNA polymerase, which is stalled at the lesion site. Then, the TCR-specific factors bind to stalled RNA polymerase, such as CSA, CSB, and XAB2 [118]. These factors initiate the assembly of other factors that result in the displacement of the polymerase complex and induces chromatin modifications that allow the exposure of lesions for further processing [119]. After DNA damage recognition a stable pre-incision complex is formed around the lesion. The multiprotein TFIIH (Transcription Factor IIH) complex is recruited to the lesion. This complex comprises of two sub-complexes: (1)–XPB DNA helicase, p62, p52, p44, p34, and p8; (2) CDK7 kinase, cyclin H, and MAT1 assembly factor. The subcomplexes are linked by XPD DNA helicase. The activity of the TFIIH complex induces the unwinding of the DNA helix, which leads to the recruitment of XPA, RPA, and XPG endonuclease. The next step is the excision of a short DNA fragment containing damaged nucleotide with the use of specific enzymes—the incision 5′ to the lesion is catalyzed by XPF/ERCC1 endonuclease that interacts with XPA and incision 3′ to the lesion by XPG endonuclease. Then, the DNA synthesis may be performed by three various polymerases (δ, ε, or κ), and DNA ligation is accomplished by Lig I or Lig III. In plants, the homologs of NER genes were identified and confirmed to be involved in this mechanism of DNA repair. For example, Arabidopsis plants defective in DDB1A, DDB1B, DDB2, CSA, and XPD are characterized by altered UV sensitivity/tolerance [120,121,122,123,124,125]. The list of NER-related proteins in Arabidopsis is provided in Table 2. The UV-induced photoadducts may be also repaired directly, not by NER, through the photoreactivation (light-dependent process), which is performed by photolyases that require light (360–420 nm) to be active [126]. There are two types of these repair enzymes in plants: Class II CPD photolyase and (6-4) photolyase [127,128].

As its name indicates, the MMR (mismatch repair) mechanism is used to fix not only mismatches but also small insertion/deletion loops (IDLs) that arise from the incorrect incorporation of nucleotides or accidental insertion/deletion of nucleotides during DNA replication in the S phase. The first step of this mechanism is the recognition of lesions by MutS complexes that have DNA binding activities. In general, in eukaryotes, mismatches are detected by specific MutSα (MSH2/MSH6 heterodimer) and MutSβ (MSH2/MSH3 heterodimer) complexes. However, plants possess an additional complex involved in this step—MutSγ (MSH2/MSH7 heterodimer) [129,130]. MutSα recognizes single-base mismatches (including oxidative and methylated mispairs) and very short IDLs, MutSβ recognizes longer IDLs (with up to 16 unpaired nucleotides), and MutSγ detects few types of single-base mismatches in plants [131,132,133]. Moreover, MutSα can also recognize UV-induced lesions such as CPD and 6-4 PPs and initiate MMR for their repair; however, these types of lesions are usually repaired through NER mechanisms [134,135]. MSH2 protein, which is a component of each known MutS complex involved in MMR, has the ability to activate ATR kinase [135,136]. The MutS complexes recruit the MutLα complex to the lesion site in an ATP-dependent manner. MutLα complex has endonuclease activity and it initiates a repair reaction by nicking the daughter strand 5′ to the mismatch. In humans, the MutLα complex is a heterodimer of MLH1 and PMS2 proteins, whereas in plants this heterodimer is composed of MLH1 and PMS1. MutLα recruits Exo1 to the nick. Exo1 is the exonuclease that conducts 5′ → 3′ excision in the daughter strand. When the excision reaches the mismatch, the MutS has to dissociate from the lesion site to allow Exo1 to continue excision [137]. Interestingly, there is strong evidence that Exo1-independent subpathway of MMR also exists, where DNA2 protein with helicase and nuclease domains plays an important role [138,139,140]. Arabidopsis homolog of human and yeast DNA2-JHS1 (JING HE SHENG 1) was confirmed to be involved in DNA repair, cell cycle regulation, and meristem maintenance in plants [141]. The MMR pathway is completed by a proper DNA re-synthesis by DNA Pol δ assisted by RFC, PCNA, and RPA, and then the nick is ligated by Lig 1 reviewed in [94,142]. The list of MMR-related proteins in Arabidopsis is provided in Table 3.

### 4.2. DSB Repair–Homologous Recombination (HR) and Non-Homologous End Joining (NHEJ)

The efficient functioning of HR and NHEJ pathways is of great importance because they are used to repair DSBs, the most harmful of all lesions. The basic principles of these two repair mechanisms are conserved among all eukaryotes.

The first step of DSB repair through HR is the resection of DNA ends, which is mediated by the MRN complex (MRE11-RAD50-NBS1) recruited at the DSB site [27,143]. MRN complex is known to bind to DNA ends and preserve them in close proximity to each other. It also recruits and activates ATM and induces excision of 5′ ends to form long single-stranded 3′ overhangs that are coated with RPA to protect them from exonucleolytic degradation [144]. In the case of DSB repair through homologous recombination, another DNA molecule (donor molecule) is needed as a repair template to recover genetic information. Its sequence should be identical or nearly identical to the damaged sequence. Depending on the cell phase it might come from a sister chromatid or homologous chromosome. To initiate homologous recombination, the BRCA1/2 (Breast Cancer 1/2) interacts with and recruits RAD51 recombinase that replaces RPA coating ssDNA. RAD51 initiates homology search and facilitates strand invasion into the homologous template. After recognition of the donor molecule, the 3′ overhang invades the dsDNA template by displacing the noncomplementary template strand and base pair with the other template strand, forming the opened structure called “D-loop” (displacement loop) [145]. In this structure, the DNA polymerase can start the elongation of the free 3′ end of the overhang, based on the homologous donor locus as a template. As the elongation proceeds, the D-loop may be enlarged by a DNA helicase or migrates together with the polymerase along the donor molecule [146]. There are two main models of DSB repair through HR: (1) DSBR–double-strand break repair, the classical HR model, which involves the formation of double Holliday junction (dHJ) intermediates; and (2) SDSA–synthesis-dependent strand annealing. (1) DSBR model may result in different end products (crossover and non-crossover) and it occurs mainly during meiosis when a non-sister chromatid from a homologous chromosome is used as a donor template. In the process of DSBR, two strands of the donor molecule are simultaneously used as templates for the elongation of two 3′ overhangs [147]. It leads to the creation of the double Holliday junction, the structure containing two interconnected DNA molecules. The dHJ intermediates are resolved (separated into two double-strand molecules) by specific endonucleases called resolvases, which may cut both crossed and non-crossed strands, resulting in the crossover or non-crossover products, respectively [2]. (2) In the case of the SDSA model, only non-crossover products arise. It is based on the elongation of only one 3′ overhang with the use of the donor strand. The elongation proceeds until it is possible to realign with the other side of the DSB in the originally damaged DNA molecule [144,148]. These two models of HR are flawless and do not lead to any loss of DNA sequence. There is also a third model of DSB repair by HR called SSA–single strand annealing. SSA can be used to repair DSBs localized between tandemly repeated sequences. The homologous repeats are used to bridge DSB ends. SSA is not flawless as it leads to a deletion rearrangement between the repeats [149]. The list of HR-related proteins in plants is provided in Table 4.

Contrary to HR, in the NHEJ pathway, the re-joining of broken DNA strands is straightforward and does not need any donor template. Because the sequence information is not used in this repair mechanism, it is error-prone [27]. There are two subpathways of NHEJ: cNHEJ (canonical NHEJ) and aNHEJ (alternative NHEJ) [150]. In the cNHEJ pathway, the DSB is recognized by Ku70/Ku80 heterodimer that forms a ring that keeps the DNA ends in close proximity and protects them from degradation. The binding of Ku70/Ku80 initiates the NHEJ repair [151]. Then, several factors involved in the resection and processing of DNA ends are recruited at the DSBs site such as PARP (POLY ADP-RIBOSE POLYMERASE) proteins, SNM1 (SENSITIVE TO NITROGEN MUSTARD), ZDP (ZINC 4 FINGER DNA 3′-PHOSPHOESTERASE), Pol λ, Rad9, and MRN complex. In the end, the complex consisting of Ligase 4 and XRCC4 performs the final ligation [152]. In the aNHEJ subpathway, which is activated in the absence of cNHEJ factors, the 3′-resection of the broken ends occurs by various exonucleases, making it similar to the SSA mechanism. When produced 3′ overhangs from both sides of DSBs have some complementary nucleotides, they can simply anneal to each other. After trimming the remaining ends, re-ligation occurs. However, the precise molecular mechanisms underlying aNHEJ in plants remain unclear [153]. The list of NHEJ-related proteins in plants is provided in Table 5. In general, in somatic cells, the NHEJ pathway plays a major role in the repair of plant DSBs, however, the HR pathway may be also widely involved in DSB repair in dividing cells, during the S and G2 phases of the cell cycle, due to the availability of sister chromatids [134]. During meiosis, where the formation of DSBs is highly controlled, the HR mechanism (precisely DSBR) is the main mode of their repair, where RAD51 acts together with meiosis specific DMC1 protein (DISRUPTED MEIOTIC CDNA1) and SMC5/6 (STRUCTURAL MAINTENANCE OF CHROMOSOME 5/6) complex [154,155].

Obviously, the DSB repair mechanisms in plants are not as well described as in humans. However, recently the DNA repair mechanisms became of special interest in plant studies because they are used in genetic engineering to achieve controlled modification of plant genomes (gene editing, i.e., through CRISPR/Cas-based methods) [152,156,157,158]. Most of the HR and NHEJ proteins have been identified and characterized in plants, and many plant mutants, especially Arabidopsis, have been studied, showing high evolutionary conservation of these mechanisms between plants and animals reviewed in [94].

Interestingly, studies performed on *Arabidopsis thaliana* established an important role for specific small RNAs (called diRNAs for DSB-induced small RNAs) in efficient DSB repair. DiRNAs are produced from sequences flanking DSB. The proposed role of diRNAs in DSB repair is that they are guide molecules directing chromatin modification and involved in recruitment of proper complexes, such as SMC5/6, a chromosomal ATPase involved in DSB repair, to DSBs. However, their precise role remains to be elucidated [159,160]. SMC5/6 complex is known to play a role in the maintenance of genome integrity; however, its mode of function is not fully understood yet [161]. It has already been revealed that several factors are involved in the recruitment of SMC5/6 to DSB site. Recently, it has been shown that SWI3B subunit of SWI/SNF (SWITCH/SUCROSE NON- FERMENTABLE) chromatin remodeling complex enhances dissociation of SMC5 from chromosomes for its further recruitment at DSB site [162].

## 5. Cell Cycle Stoppage—Giving the Cell Time to Repair

The main regulator of the DDR pathway in plants, SOG1, activates directly and indirectly not only many genes related to DNA repair, but also those related to cell cycle regulation. Dividing tissues are the most sensitive to DNA-damaging agents. Cell cycle arrest is the first effect of DDR activation and is crucial to allow time to repair to avoid transmission of lesions to daughter cells. In general, the cell can be arrested in the G2/M or S phase of the cell cycle (by activation of G2/M and replication checkpoints, respectively). The arrest depends on the phase at which DNA damage occurs [60].

The progression of the cell cycle (through G1-S-G2-M phases) in plants relies on an enormous number of cell cycle regulators, very often encoded by multiple loci. In principle, the cell cycle is controlled by various cyclin-dependent kinases (CDKs) that are active only when properly phosphorylated and in complex with appropriate cyclins. Their activity is highly increased at the transition from G1 to S and from G2 to M. At the G1/S transition, they phosphorylate many proteins important for DNA synthesis, whereas at the G2/M transition, they phosphorylate proteins related to chromosome segregation. There are many mechanisms modulating CDK-cyclin activity, such as transcriptional regulation, protein degradation, phosphorylation of Thr residues, and binding of specific inhibitors [163,164].

In response to DNA damage, SOG1 activates the transcription of genes encoding inhibitors of cyclin-dependent kinases. Two of them, *SMR5* and *SMR7* (*Siamese Related 5/7*), are confirmed to be direct targets of SOG1 in Arabidopsis [87]. Upon DNA stress, SOG1 represses the transcription of specific B-type CDKs—CDKB2s that are mitotic regulators in Arabidopsis [83,165]. This suppression leads to the activation of the G2/M checkpoint and blockage of the cell cycle. On the other hand, SOG1 enhances the activity of plant-specific CDKB1 by stimulation of CYCB1 (which forms a complex with CDKB1 and is widely used as a marker for cell proliferation). This conundrum was solved by confirming that CDKB1-CYCB1 also mediates homologous recombination (HR), showing the dual face of cyclin B1 [21,166].

SOG1 activates the G2/M checkpoint also via stimulation of some *MYB3R* genes—*MYB3R1*, *MYB3R3*, and *MYB3R5* (collectively called *Rep-MYB3R*)—that encode major repressors of genes related to the regulation of G2/M transition and are required for M phase onset [86]. Principally, CDK complexes phosphorylate Rep-MYB3R and directs them to proteosomal degradation, which allows G2/M progression. However, under genotoxic stress, CDK activity is inhibited by SOG1 (via activation of *SMR5* and *SMR7*), which leads to the accumulation of Rep-MYB3R—this halts G2/M progression [32]. Recent studies in Arabidopsis have shown that two NAC transcription factors closely related to SOG1, ANAC044 and ANAC085 (that are also regulated by SOG1), are involved in Rep-MYB3R stabilization and accumulation, and hence the activation of the G2/M checkpoint [167]. Additionally, ANAC044, together with homologs of the human DREAM (DIMERIZATION PARTNER, RB-LIKE, E2F AND MULTI-VULVAL CLASS B) complex, was found to be a part of the RBR1 interactome. DREAM complex is very well studied in humans as a repressive regulator of the cell cycle; however, in plants, the full assembly of this complex was not known until recently. The newest data indicate the existence of multiple DREAM complexes in plants that mediate growth arrest upon DNA stress in conjunction with ANAC044 [168].

The existence of DNA lesions may perturb the course of DNA replication (replication stress), causing the cell cycle arrest in the S phase. The evolutionary conserved WEE1 kinase is a key Intra-S checkpoint protein [164]. It is confirmed in Arabidopsis that this kinase is accumulated in S-phase nuclei upon replication stress [169]. The WEE1 kinase, regulated by both SOG1 and RBR1/E2FA complex, is negatively controlling the activity of CDKs by their inhibitory phosphorylation (at Thr-14 and Tyr-15 positions), which leads to delay of S-phase progression [85,170]. Recently, it has been suggested that WEE1 can induce cell cycle arrest by phosphorylation of other proteins, e.g., FBL17 (F-box-like 17). FBL17 is an E3 ubiquitin ligase that promotes the degradation of CDK inhibitors. In Arabidopsis, phosphorylation of FBL17, driven by WEE1, directs it to degradation and hence leads to the accumulation of CDK inhibitors [171]. A novel cell cycle control mechanism regulated by WEE1 has been lately revealed in Arabidopsis by Wang et al. During replication stress, WEE1 was found to directly phosphorylate PRL1-the core protein of MAC (MOS4-associated complex) involved in alternative splicing. This phosphorylation leads to proteosomal degradation of PRL1, which in consequence induces intron retention of cell cycle genes (e.g., *CYCD1;1* and *CYCD3;1*) contributing to cell cycle arrest in the S phase [172]. Such a delay of S-phase progression empowers DNA repair during replication stress.

## 6. Endoreduplication and Programmed Cell Death

In the case of an extreme number of DNA lesions, when DNA repair machinery is not able to fix them, the endoreduplication (endoreplication) may be activated. In the regular cell cycle, one DNA replication round is followed by mitotic division. Endoreduplication is a situation when the cell is replicating its DNA several times without the following mitosis, which causes an increase in ploidy level and usually leads to enlargement and differentiation of the cell [173]. Endoreduplication is known to be implicated in various stress responses in plants [174]. In general, the arrest of the cell cycle in the G2 phase triggers the transition from mitotic division to endoreduplication [175]. One of the most important factors required for the progression of endoreduplication in plants is the DNA topoisomerase VI complex, that in Arabidopsis is composed of AtSPO11-3 (one of the homologs of TOPOISOMERASE VIA), AtTOP6B (TOPOISOMERASE 6 SB), RHL1 (ROOT HAIRLESS 1), and MIDGET proteins [176,177,178]. The programmed induction of endoreduplication in response to DSBs in Arabidopsis helps to prevent the transmission of damaged DNA to daughter cells through proliferation, but, on the other hand, it maintains growth [31]. However, in some plant species, such as rice (*Oryza sativa*), endoreduplication does not occur as a response to extreme DNA damage [179].

In order to erase damaged cells, in the case of severe genotoxic stress, programmed cell death (PCD) may be activated. It is a genetically regulated biochemical pathway of organized cell destruction [180]. It is an essential element of normal plant development [181], but additionally, it may be activated upon different stresses, both biotic and abiotic, including genotoxic stress [182]. In humans, PCD induced by DNA stress is governed by p53 and caspase cascade. Caspases are cysteine proteases that cleave a set of target proteins causing disassembly of the cell [183]. However, in plants, there are no homologs of either p53 or caspases. SOG1, as a functional homolog of p53, is the key player in PCD in response to severe genotoxic stress, together with its two targets—also NAC transcription factors—ANAC044 and ANAC085 [167]. However, the downstream PCD effectors are not yet fully elucidated in plants [182].

## 7. Concluding Remarks

The proper functioning of the DDR pathway is crucial to preserve genome integrity. It is responsible for orchestrating the processes of DNA repair, cell cycle stoppage, and cell death upon genotoxic stress that may be caused by many factors, including environmental ones. Our understanding of plant DDR has greatly increased over recent years; however, it still lags behind that of animal and human DDR. Most of the knowledge collected for plants concerns a model species *Arabidopsis thaliana*, and the new challenge is to reveal the details of these processes in crops. Plants exposed to DNA-damaging conditions display a significant reduction in productivity and yield, which can have a huge agronomic and economic impact. Detailed knowledge about maintaining the DDR pathway in crops may be useful in the breeding of new cultivars more tolerant to stresses causing DNA damage.

## Figures and Tables

**Figure 1 ijms-24-02404-f001:**
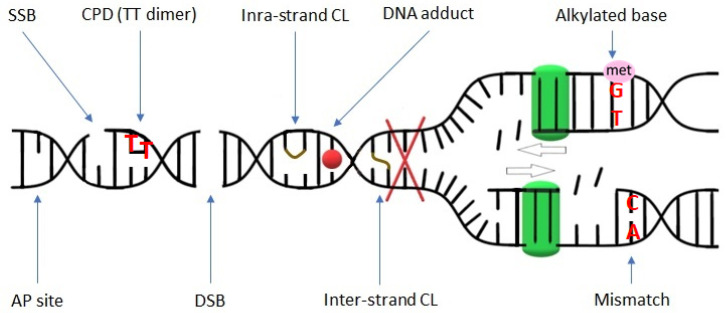
Schematic representation of various DNA lesions. Red X indicates the blockage of the replication fork (replication stress) that may be caused by DNA damage. The green squares symbolize polymerase performing replication. SSB—single-strand break; CPD—cyclobutene pyrimidine dimer; CL—crosslink; AP site—apurinic/apyrimidinic site; DSB—double-strand break.

**Figure 2 ijms-24-02404-f002:**
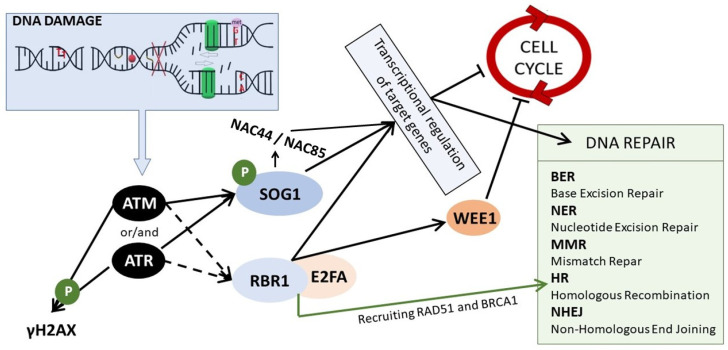
The scheme of DDR signaling in plants. In response to DNA damage, ATM and/or ATR kinases are activated. ATM is activated upon DSBs, whereas ATR upon SSBs and replication stress. They phosphorylate and activate SOG1, which regulates genes responsible for cell cycle stoppage and DNA repair. This regulation is both direct and indirect, i.e., through other NAC TFs, such as NAC44 and NAC85. ATM and ATR kinases lead also to the phosphorylation of histone H2AX, and they induce the assembly of the RBR1/E2FA complex that regulates the expression of genes related to the cell cycle arrest and is also involved in the DNA repair through HR by recruiting RAD51 and BRCA1 to the DNA lesions.

**Table 1 ijms-24-02404-t001:** BER-related proteins in *Arabidopsis thaliana*.

Protein	Full Name	Acc. No.	Function
FPG	FORMAMIDOPYRIMIDINE-DNA GLYCOSYLASE	At1g52500	Bifunctional glycosylases (glycosylase and AP lyase activity)
OGG1	8-OXOGUANINE-DNA GLYCOSYLASE1	At1g21710	
NTH1	ENDONUCLEASE III 1	At2g31450	
NTH2	ENDONUCLEASE III 2	At1g05900	
DME	DEMETER	At5g04560	
DML2	DEMETER-LIKE PROTEIN 2	At3g10010	
DML3	DEMETER-LIKE PROTEIN 3	At4g34060	
ROS1	REPRESSOR OF SILENCING 1	At2g36490	
UNG (UDG)	URACIL DNA GLYCOSYLASE	At3g18630	Monofunctional glycosylases
AAG	ALKYLADENINE-DNA GLYCOSYLASE	At3g12040	
MBD4L	METHYL-CPG-BINDING DOMAIN PROTEIN 4 LIKE	At3g07930	
ARP	APURINIC ENDONUCLEASE-REDOX PROTEIN	At2g41460	AP endonucleases
APE1L	APURINIC/APYRIMIDINIC ENDONUCLEASE1-LIKE PROTEIN	At3g48425	
APE2	APURINIC/APYRIMIDINIC ENDONUCLEASE2	At4g36050	
ZDP	ZINC 4 FINGER DNA 3′-PHOSPHOESTERASE	At3g14890	3′ DNA phosphatase
XRCC1	HOMOLOG OF X-RAY REPAIR CROSS COMPLEMENTING 1	At1g80420	Stimulation of phosphatase activity of ZDP and enhancement of DNA ligation by interaction with LIG1
FEN1 (SAV6)	FLAP ENDONUCLEASE I	At5g26680	Flap endonuclease
Pol δ	DNA POLYMERASE DELTA	At1g09815 At2g42120 At5g63960	Polymerases
Pol ε	DNA POLYMERASE EPSILON	At1g08260 At2g27120 At5g22110	
Pol λ	DNA POLYMERASE LAMBDA	At1g10520	
LIG1	LIGASE 1	At1g08130	Ligase involved in the final ligation of DNA ends in SP and LP-BER
LIG4	LIGASE 4	At5g57160	Ligases critical for seed longevity
LIG6	LIGASE 6	At1g66730	

**Table 2 ijms-24-02404-t002:** NER-related proteins in *Arabidopsis thaliana*.

Protein		Full Name	Acc. No.	Function
XPC-HR23B-CEN2 complex	RAD4 (XPC homolog)	RADIATION SENSITIVE PROTEIN 4	At5g16630	Recognition of lesions in GG-NER (Global Genomic Repair–NER)
	RAD23A	RADIATION SENSITIVE PROTEIN 23A	At1g16190	
	RAD23B	RADIATION SENSITIVE PROTEIN 23B	At1g79650	
	RAD23C	RADIATION SENSITIVE PROTEIN 23C	At3g02540	
	RAD23D	RADIATION SENSITIVE PROTEIN 23D	At5g38470	
	CEN2	CENTRIN 2	At4g37010	
DDB complex	DDB1A	DAMAGED DNA-BINDING PROTEIN 1A	At4g05420	
	DDB1B	DAMAGED DNA-BINDING PROTEIN 1B	At4g21100	
	DDB2	DAMAGED DNA-BINDING PROTEIN 2	At5g58760	
CSA1		COCKAYNE SYNDROME GROUP A 1	At1g27840	Recognition of lesions in TC-NER (Transcriptional-Coupled Repair–NER)
CSA2		COCKAYNE SYNDROME GROUP A 2	At1g19750	
CHR8 (CSB homolog)		CHROMATIN REMODELING 8	At2g18760	
CHR24 (CSB homolog)		CHROMATIN REMODELING 24	At5g63950	
Preincision complex	XPB1	XERODERMA PIGMENTOSUM GROUP B 1	At5g41370	TFIIH core complex with helicase activity
	XPB2	XERODERMA PIGMENTOSUM GROUP B 2	At5g41360	
	XPD (UVH6)	XERODERMA PIGMENTOSUM GROUP D (ULTRAVIOLET HYPERSENSITIVE 6)	At1g03190	
	TTDA	TRICHOTHIODYSTROPHY GROUP A	At1g12400 At1g62886	
	GTF2H2	GENERAL TRANSCRIPTION FACTOR II H2	At1g05055	
	At1g18340		At1g18340	
	At1g55750		At1g55750	
	At3g61420		At3g61420	
	At4g17020		At4g17020	
	TTDA	TRICHOTHIODYSTROPHY GROUP A	At1g12400	
			At1g62886	
	CDKD1;1	CYCLIN-DEPENDENT KINASE D1; 1	At1g73690	Module of THIIH with kinase activity
	CDKD1;2	CYCLIN-DEPENDENT KINASE D1; 2	At1g66750	
	CDKD1;3	CYCLIN-DEPENDENT KINASE D1; 3	At1g18040	
	CYCH;1	CYCLIN H;1	At5g27620	
	MAT1	MÉNAGE À TROIS	At4g30820	
UVH1 (XPF homolog)		ULTRAVIOLET HYPERSENSITIVE 1	At5g41150	Nucleases performing incisions 5′ or 3′ to the lesion
UVH3 (XPG homolog)		ULTRAVIOLET HYPERSENSITIVE 3	At3g28030	
ERCC1		EXCISION REPAIR CROSS-COMPLEMENTATION GROUP 1	At3g05210	
RPA (three subunits: RPA1/2/3)		REPLICATION PROTEIN A 1	Multiple loci *: At2g06510 At5g08020 At5g45400 At5g61000 At4g19130 At2g24490 At3g02920 At3g52630 At4g18590	Complex involved in the excision of the damaged nucleotide
		REPLICATION PROTEIN A 2		
		REPLICATION PROTEIN A 3		
PCNA1		PROLIFERATING CELLULAR NUCLEAR ANTIGEN 1	At1g07370	Cofactors of polymerases
PCNA2		PROLIFERATING CELL NUCLEAR ANTIGEN 2	At2g29570	
RFC (five subunits: RFC1/2/3/4/5)		REPLICATION FACTOR C	At1g21690 At1g63160 At1g77470 At5g22010 At5g27740	Complex involved in the loading of PCNAs onto the DNA strand
POL δ		DNA POLYMERASE DELTA	see Table 1	Polymerases
Pol ε		DNA POLYMERASE EPSILON	see Table 1	
LIG1		LIGASE 1	At1g08130	Ligase

* multiple genes encode the RPA subunits in *Arabidopsis thaliana*.

**Table 3 ijms-24-02404-t003:** MMR-related proteins in *Arabidopsis thaliana*.

Protein		Full Name	Acc. No.	Function
MutS complexes	MSH2	MUTS HOMOLOG 2	At3g18524	Recognition of mismatches
	MSH3	MUTS HOMOLOG 3	At4g25540	
	MSH6	MUTS HOMOLOG 6	At4g02070	
	MSH7	MUTS HOMOLOG 7	At3g24495	
MutLα complex	MLH1	MUTL HOMOLOG 1	At4g09140	Endonuclease activity
	PMS1	POSTMEIOTIC SEGREGATION 1	At4g02460	
EXO1		EXONUCLEASE 1	At1g29630	Exonuclease
RFC		REPLICATION FACTOR C	see Table 2	Complex involved in the loading of PCNAs onto the DNA strand
PCNA1		PROLIFERATING CELLULAR NUCLEAR ANTIGEN 1	At1g07370	Cofactors of polymerase
PCNA2		PROLIFERATING CELLULAR NUCLEAR ANTIGEN 2	At2g29570	
DNA2 (JHS1)		JING HE SHENG 1		Helicase and nuclease activity
RPA		REPLICATION PROTEIN A	see Table 2	Complex involved in the excision of the lesion
POL δ		DNA POLYMERASE DELTA	see Table 1	Polymerase
LIG1		LIGASE 1	At1g08130	Ligase

**Table 4 ijms-24-02404-t004:** HR-related proteins in *Arabidopsis thaliana*.

Protein		Full Name	Acc. No.	Function
MRN complex	MRE11	MEIOTIC RECOMBINATION 11	At5g54260	Bind to DNA ends and preserve them in close proximity, induce excision of 5′ end
	RAD50	RADIATION SENSITIVE PROTEIN 50	At2g31970	
	NBS1	NIJMEGEN BREAKAGE SYNDROME 1	At3g02680	
RAD51		RADIATION SENSITIVE PROTEIN 51	At5g20850	Recombinase initiating homology search and strand invasion
RPA		REPLICATION PROTEIN A	see Table 2	Coats 3′ overhangs–protection from exonucleolytic degradation
GR1/COM1		GAMMA RESPONSE 1	At3g52115	DSB end processing
EXO1		EXONUCLEASE 1	At1g29630 At1g18090	
RECQ4A RECQ4B		ATP-DEPENDENT DNA HELICASE Q-LIKE 4A AND 4B	At1g10930 At1g60930	
RAD52		RADIATION SENSITIVE PROTEIN 52	At1g71310 At5g47870	Recombination mediators
BRCA1		BREAST CANCER 1	At4g21070	
BRCA2		BREAST CANCER 2	At5g01630 At4g00020	
RAD51B		RADIATION SENSITIVE PROTEIN 51B	At2g28560	
RAD51C		RADIATION SENSITIVE PROTEIN 51C	At2g45280	
RAD51D		RADIATION SENSITIVE PROTEIN 51D	At1g07745	
XRCC2		X-RAY REPAIR CROSS COMPLEMENTING 2	At5g64520	
XRCC3		X-RAY REPAIR CROSS COMPLEMENTING 3	At5g54750	
RAD54		RADIATION SENSITIVE PROTEIN 54	At3g19210	dsDNA translocase–stimulates the D-loop formation
PCNA1		PROLIFERATING CELLULAR NUCLEAR ANTIGEN 1	At1g07370	Cofactors of polymerases
PCNA2		PROLIFERATING CELL NUCLEAR ANTIGEN 2	At2g29570	
RFC		REPLICATION FACTOR C	see Table 2	Complex involved in the loading of PCNAs onto the DNA strand
POL δ		DNA POLYMERASE DELTA	See Table 1	Polymerase
SRS2		SUPPRESSOR OF RAD SIX-SCREEN MUTANT 2	At4g25120	Helicase activity
FANCM		FANCONI ANEMIA GROUP M PROTEIN	At1g35530	
EME1		ESSENTIAL MEIOTIC ENDONUCLEASE 1	At2g21800 At2g22140	Endonuclease activity
MUS81		MMS AND UV SENSITIVE 81	At4g30870	
GEN1		GEN1	At1g01880	Holliday junction 5′ flap endonuclease
SEND1		SINGLE-STRAND DNA ENDONUCLEASE1	At3g48900	
TOP3α		TOPOISOMERASE 3 ALPHA	At5g63920	Together with RECQ4A involved in HJ dissolution
RMI1		RECQ-MEDIATED GENOME INSTABILITY PROTEIN 1	At5g63540	

**Table 5 ijms-24-02404-t005:** NHEJ-related proteins in *Arabidopsis thaliana*.

Protein	Full Name	Acc. No.	Function
KU70	KU70 HOMOLOG	At1g16970	KU70-KU80 heterodimer bind and protect DSB ends
KU80	KU80 HOMOLOG	At1g48050	
PARP	POLY ADP-RIBOSE POLYMERASE	At2g31320 At4g02390 At5g22470	DNA end binding NAD + ADP-ribosyltransferase
MRN complex (MRE11-RAD50-NBS1)	MRE11	At5g54260	Bind to DNA ends and preserve them in close proximity, induce excision of 5′end
	RAD50	At2g31970	
	NBS1	At3g02680	
Snm1	SENSITIVE TO NITROGEN MUSTARD 1	At3g26680	DNA end processing
ZDP	ZINC 4 FINGER DNA 3′-PHOSPHOESTERASE	At3g14890	
Rad9	RADIATION SENSITIVE PROTEIN 9	At3g05480	
Pol λ	POLYMERASE λ	At1g10520	Polymerase
POLθ	POLYMERASE θ	At4g32700	
LIG4	LIGASE 4	At5g57160	Ligase
XRCC4	X-RAY REPAIR CROSS COMPLEMENTING 4	At3g23100	Complex with LIG4

## Data Availability

Not applicable.

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
