# Peer review of "How Do Plants Cope with DNA Damage? A Concise Review on the DDR Pathway in Plants"

_ijms, 2023, doi:10.3390/ijms24032404_

Round 1

Reviewer 1 Report

In the present study by Szurman-Zubrzycka et al entitled How Do Plants Cope with DNA Damage? A Concise Review on the DDR Pathway in Plants, authors give very extensive and detailed overview of DNA damage responses in plants.

Although I recommend paper for publication in present form, I have however some suggestions for improvement. In my opinion abstract should be focused more on specificity of the paper. It is now focused more on general DNA damage information, so I would suggest pinpointing here differences or specificities of plant DNA damage responses. I think it is important to write in abstract what is importance of such a review. For example, in conclusion agronomic impact is nicely mentioned. It would be also worth considering writing about this agronomic impact in at least few sentences somewhere in the text.

I think also introduction should be improved in order for reader to get more insight into plant DNA damage responses and the model Arabidopsis.

Text from lines 115-126 is very interesting for plant specific DNA damage. Authors are referring to reviews, but it would be worth to make this section longer.

Author Response

"I have however some suggestions for improvement. In my opinion abstract should be focused more on specificity of the paper. It is now focused more on general DNA damage information, so I would suggest pinpointing here differences or specificities of plant DNA damage responses. I think it is important to write in abstract what is importance of such a review. For example, in conclusion agronomic impact is nicely mentioned. It would be also worth considering writing about this agronomic impact in at least few sentences somewhere in the text.

Dear reviewer,

Thank you very much for your review. We appreciate that you recommend our paper for publication. According to your indication we slightly expanded the abstract section, by indicating the example of plant-specific elements of DDR and highlighting the agronomic impact of this knowledge (its special importance for crops). 

I think also introduction should be improved in order for reader to get more insight into plant DNA damage responses and the model Arabidopsis. Text from lines 115-126 is very interesting for plant specific DNA damage. Authors are referring to reviews, but it would be worth to make this section longer.

Thank you for this suggestion. We have provided more information about DNA damage caused by cold stress, heat stress and salinity, where we refer to research articles, not only reviews (lines 136-144). We hope these amendments meet your expectations.

Reviewer 2 Report

DNA damage responses (DDR) are essential to maintain genome stability, growth and development, and stress responses. However, the mechanisms of plant DDR are far less well-understood than those in animals. In recent years, there are many important progresses on plant DDR. Since there are several review papers on plant DDR, readers can read these reviews for general and basic information on this topic. I think the authors need to put more effort to summarize the recent advances. 

1. The authors summarized some DNA-damaging agents used in plants. I suggested the authors include Methyl methanesulphonate (MMS) and camptothecin (CPT). 

2. Figure 2 is similar to figures in other review papers. I suggested the authors draw a more detailed model including the newly identified regulators. 

3. I suggested the authors include the following important references to reflect recent advances.

(1) Chen, H., He, C., Wang, C., Wang, X., Ruan, F., Yan, J., Yin, P., Wang, Y. and Yan, S., 2021. RAD51 supports DMC1 by inhibiting the SMC5/6 complex during meiosis. The Plant Cell, 33(8), pp.2869-2882.

(2) Davarinejad, H., Huang, Y.C., Mermaz, B., LeBlanc, C., Poulet, A., Thomson, G., Joly, V., Muñoz, M., Arvanitis-Vigneault, A., Valsakumar, D. and Villarino, G., 2022. The histone H3. 1 variant regulates TONSOKU-mediated DNA repair during replication. Science, 375(6586), pp.1281-1286.

(3) Fan, T., Kang, H., Wu, D., Zhu, X., Huang, L., Wu, J. and Zhu, Y., 2022. Arabidopsis γ-H2A. X-INTERACTING PROTEIN participates in DNA damage response and safeguards chromatin stability. Nature Communications, 13(1), pp.1-14.

(4) Jiang, J., Mao, N., Hu, H., Tang, J., Han, D., Liu, S., Wu, Q., Liu, Y., Peng, C., Lai, J. and Yang, C., 2019. A SWI/SNF subunit regulates chromosomal dissociation of structural maintenance complex 5 during DNA repair in plant cells. Proceedings of the National Academy of Sciences, 116(30), pp.15288-15296.

(5) Jiang, J., Ou, X., Han, D., He, Z., Liu, S., Mao, N., Zhang, Z., Peng, C.L., Lai, J. and Yang, C., 2022. A diRNA–protein scaffold module mediates SMC5/6 recruitment in plant DNA repair. The Plant Cell, 34(10), pp.3899-3914.

(6) (15)Lang, L., Pettkó-Szandtner, A., Elbaşı, H.T., Takatsuka, H., Nomoto, Y., Zaki, A., Dorokhov, S., De Jaeger, G., Eeckhout, D., Ito, M. and Magyar, Z., 2021. The DREAM complex represses growth in response to DNA damage in Arabidopsis. Life science alliance, 4(12).

(7) Li, J., Liang, W., Liu, Y., Ren, Z., Ci, D., Chang, J. and Qian, W., 2022. The Arabidopsis ATR-SOG1 signaling module regulates pleiotropic developmental adjustments in response to 3'-blocked DNA repair intermediates. The Plant Cell, 34(2), pp.852-866.

(8) Li, J., Liang, W., Li, Y. and Qian, W., 2018. Apurinic/apyrimidinic endonuclease2 and zinc finger dna 3′-phosphoesterase play overlapping roles in the maintenance of epigenome and genome stability. The Plant Cell, 30(9), pp.1954-1970.

(9) Pan, T., Gao, S., Cui, X., Wang, L. and Yan, S., 2022. APC/CCDC20 targets SCFFBL17 to activate replication stress responses in Arabidopsis. The Plant Cell.

(10) Pedroza-Garcia, J.A., Eekhout, T., Achon, I., Nisa, M.U., Coussens, G., Vercauteren, I., Van den Daele, H., Pauwels, L., Van Lijsebettens, M., Raynaud, C. and De Veylder, L., 2021. Maize ATR safeguards genome stability during kernel development to prevent early endosperm endocycle onset and cell death. The Plant Cell, 33(8), pp.2662-2684.

(11) Prochazkova, K., Finke, A., Tomaštíková, E.D., Filo, J., Bente, H., Dvořák, P., Ovečka, M., Šamaj, J. and Pecinka, A., 2022. Zebularine induces enzymatic DNA–protein crosslinks in 45S rDNA heterochromatin of Arabidopsis nuclei. Nucleic acids research, 50(1), pp.244-258.

(12) Wang, L., Chen, H., Wang, C., Hu, Z. and Yan, S., 2018. Negative regulator of E2F transcription factors links cell cycle checkpoint and DNA damage repair. Proceedings of the National Academy of Sciences, 115(16), pp.E3837-E3845

(13) Wang, X., Wang, L., Huang, Y., Deng, Z., Li, C., Zhang, J., Zheng, M. and Yan, S., 2022. A plant-specific module for homologous recombination repair. Proceedings of the National Academy of Sciences, 119(16), p.e2202970119.

(14) Wei, W., Ba, Z., Gao, M., Wu, Y., Ma, Y., Amiard, S., White, C., Danielsen, J., Yang, Y., and Qi, Y., 2012. A role for small RNAs in DNA double-strand break repair. Cell, 149(1), pp.101-112.  

(15) Willems, A., Heyman, J., Eekhout, T., Achon, I., Pedroza-Garcia, J.A., Zhu, T., Li, L., Vercauteren, I., Van den Daele, H., Van De Cotte, B. and De Smet, I., 2020. The cyclin CYCA3; 4 is a postprophase target of the APC/CCCS52A2 E3-ligase controlling formative cell divisions in Arabidopsis. Plant Cell, 32(9), pp.2979-2996.

Author Response

Dear Reviewer,

Thank you very much for your review. We very appreciate it. Below we refer to the three points raised by you:

  1. The authors summarized some DNA-damaging agents used in plants. I suggested the authors include Methyl methanesulphonate (MMS) and camptothecin (CPT). 

Thank you for this indication – we have included MMS in the paragraph regarding alkylating agents (lines 69-70) and we have added information about camptothecin (lines 99-105), as well as  information regarding zebularine (using reference suggested by reviewer)(lines 105-108).

  1. Figure 2 is similar to figures in other review papers. I suggested the authors draw a more detailed model including the newly identified regulators. 

According to your suggestion, we have included in the Figure 2 the recently confirmed regulators known to act in DDR in plants – transcription factors from the NAC family – NAC44 and NAC85. We intended our figure to be simple, to make it easy to understand. more details of the regulatory network are provided in the text of the manuscript. The respective description of the Figure has also been changed.

  1. I suggested the authors include the following important references to reflect recent advances.

Thank you for suggesting a list of references that can enhance our manuscript. We have added most of them that we found relevant and matching to our manuscript. We believe that it improved our article.

- Chen, H., He, C., Wang, C., Wang, X., Ruan, F., Yan, J., Yin, P., Wang, Y. and Yan, S., 2021. RAD51 supports DMC1 by inhibiting the SMC5/6 complex during meiosis. The Plant Cell, 33(8), pp.2869-2882.

We have added information about DMC1 and SMC5/6 complex when mentioning HR during meiosis (lines 442-444)

- Fan, T., Kang, H., Wu, D., Zhu, X., Huang, L., Wu, J. and Zhu, Y., 2022. Arabidopsis γ-H2A. X-INTERACTING PROTEIN participates in DNA damage response and safeguards chromatin stability. Nature Communications, 13(1), pp.1-14.

We have added information about XIP protein to the „DDR – sensing and signaling the DNA damage” section (lines 188-190)

- Lang, L., Pettkó-Szandtner, A., Elbaşı, H.T., Takatsuka, H., Nomoto, Y., Zaki, A., Dorokhov, S., De Jaeger, G., Eeckhout, D., Ito, M. and Magyar, Z., 2021. The DREAM complex represses growth in response to DNA damage in Arabidopsis. Life science alliance, 4(12).

This article has already been cited in the manuscript.

- Li, J., Liang, W., Li, Y. and Qian, W., 2018. Apurinic/apyrimidinic endonuclease2 and zinc finger dna 3′-phosphoesterase play overlapping roles in the maintenance of epigenome and genome stability. The Plant Cell, 30(9), pp.1954-1970.

We have added information about overlapping roles af APE2 and ZDP phosphates in BER in Arabidopsis (lines 283-285)

-Prochazkova, K., Finke, A., Tomaštíková, E.D., Filo, J., Bente, H., Dvořák, P., Ovečka, M., Šamaj, J. and Pecinka, A., 2022. Zebularine induces enzymatic DNA–protein crosslinks in 45S rDNA heterochromatin of Arabidopsis nuclei. Nucleic acids research, 50(1), pp.244-258.

As indicated above, we have added the information about zebularine to the manuscript (lines 105-108).

- Wei, W., Ba, Z., Gao, M., Wu, Y., Ma, Y., Amiard, S., White, C., Danielsen, J., Yang, Y., and Qi, Y., 2012. A role for small RNAs in DNA double-strand break repair. Cell, 149(1), pp.101-112.  

- Jiang, J., Ou, X., Han, D., He, Z., Liu, S., Mao, N., Zhang, Z., Peng, C.L., Lai, J. and Yang, C., 2022. A diRNA–protein scaffold module mediates SMC5/6 recruitment in plant DNA repair. The Plant Cell, 34(10), pp.3899-3914.

- Jiang, J., Mao, N., Hu, H., Tang, J., Han, D., Liu, S., Wu, Q., Liu, Y., Peng, C., Lai, J. and Yang, C., 2019. A SWI/SNF subunit regulates chromosomal dissociation of structural maintenance complex 5 during DNA repair in plant cells. Proceedings of the National Academy of Sciences, 116(30), pp.15288-15296.

We have added information about SMC5/6 involment in DSBs repair and its recruitment to DSB site by diRNAs and SWI3B subunit of SWI/SNF chromatin remodeling complex and cited the three articles indicated above (lines 456-467).

According to the suggestion of both reviewers we have broaden some parts of manuscript. In total the number of references increased by 24.

We hope that our amendments meet your expectations.